# Siamese Fighting Fish (*Betta splendens* Regan) Gut Microbiota Associated with Age and Gender

Lucsame Gruneck [1,†] , Vasana Jinatham [1,2,†] , Phatthanaphong Therdtatha [3,†] and Siam Popluechai [1,2,*]

1 Gut Microbiome Research Group, Mae Fah Luang University, Muang, Chiang Rai 57100, Thailand
2 School of Science, Mae Fah Luang University, Muang, Chiang Rai 57100, Thailand
3 Division of Biotechnology, School of Agro-Industry, Faculty of Agro-Industry, Chiang Mai University, Chiang Mai 50100, Thailand
* Correspondence: siam@mfu.ac.th
† These authors contributed equally to this work.

**Abstract:** The Siamese fighting fish (*Betta splendens* Regan) is a popular ornamental fish in Thailand. Growing research suggests that fish health is influenced by gut microbiota. Here, we investigated, for the first time, the differences in the gut microbiota profiles of healthy Siamese fighting fish during the young (8-week-old) and adult male and female (16-week-old and 24-week-old) life stages using 16S rRNA gene sequence analysis. The fish were raised in controlled water quality conditions and fed on *Moina macrocopa*. Profiling of gut microbial communities revealed significant differences in the overall bacterial profile between young and adult Siamese fighting fish. Bacterial diversity decreased in the 24-week-old adult fish. Proteobacteria was the most predominant bacterial phylum in the gut of both young and adult carnivorous betta fish, in which the abundance of its members varied with age. *Plesiomonas* was enriched in male fish aged 24 weeks, whereas *Pseudomonas* dominated the gut of 8-week-old fish. Moreover, differences in predicted functions of these Proteobacteria between the young and adult fish could be a key target for improving fish growth. These findings expand our understanding of the role of gut microbiota and its association with host factors.

**Keywords:** Siamese fighting fish; gut microbiota; 16S rRNA sequencing; betta fish

## 1. Introduction

The Siamese fighting fish (*Betta splendens* Regan) is a popular ornamental fish of high economic value. This freshwater fish belongs to the family *Osphronemidae*, including 55 different Betta species recorded in Southeast Asia [1]. Of these, ten wild-type species are normally found in Thailand, where *B. splendens* Regan being the most common [1]. It is also regarded as one of Thailand's most important export ornamental fish species, with the United States, Japan, and Europe (particularly Germany, France, and the United Kingdom), as the primary markets [2]. Thailand has recently exported over 23 million betta fish, with an official value of 4.15 million USD annually, and the export trend keeps increasing [3]. However, this species, is vulnerable to changes in their aquatic environment, which may endanger their survival in nature [4]. Raising fish for high-quality mass export could be challenging, even in a controlled system due to disease encounters and fish health maintenance.

The microbial community in the gastrointestinal (GI) tract, known as the gut microbiota, in aquatic animals is complex and constitutes several aerobic and anaerobic microbes. Microbiota, mainly bacteria, begin to colonize the fish gut in the larval stage and become more complex in adults, accounting for $10^8$ microbial cells with over 500 different species [5,6]. The water environment is indispensable for fish growth in aquaculture. It is also an ideal medium for microbial growth, which colonizes in the fish gut denser than in the terrestrial animal gut [7]. Microbiota can be either autochthonous, which colonizes the fish gut, or allochthonous, which are transient free-living microbes [8]. Facultative

anaerobic bacteria, such as *Actinobacter*, *Aeromonas*, *Flavobacterium*, *Lactococcus*, *Pseudomonas*, *Bacteroides*, *Clostridium*, *Fusobacterium*, and the members of *Enterobacteriaceae* are mostly dominant in the gut of freshwater fish [9]. While the host provides an environment conducive to developing of these microbes, their relationship with the fish could affect its host functions (e.g., metabolism, immune system) [10].

According to previous studies [1,11], the stage and development of betta fish could be divided into four stages, including larval (24–48 h), juvenile (2–3 weeks), young (7–10 weeks), and adult (after 3–5 months). Adult male and female betta fish can be morphologically distinguished by color pattern and body size. It has been shown that host characteristics (such as age, sex, and sexual development) impact the composition of the fish gut microbiota, with these factors distinguishing the bacterial profiles between young and adult fish [12,13]. For example, shifts in the microbial profile were previously evidenced in zebrafish throughout its development, in which variations increased as the host aged [14]. This underlines the important role of host factors in shaping the gut microbiome. However, it should be noted that no such studies have been conducted to explore the effects of age and sex on gut microbiota in betta fish. This suggests that understanding the influence of betta fish development on microbiota colonization would contribute to bridging this gap and provide the foundation for future species conservation and fish health improvement.

In this pilot study, we used high-throughput sequencing of the 16S rRNA genes to characterize the gut microbiome in Thai betta fish of different ages and genders. *B. splendens* Regan was selected as a betta fish representative because it is a well-known and widespread *Betta* species in Thailand. We aimed to establish the community profiles and functional involvement of *B. splendens* Regan (hereafter referred to as Siamese fighting fish) gut microbiota, which may serve as baseline data for future research, such as improving the survival rate of this economically important fish species in aquaculture by manipulating their gut microbiome.

## 2. Materials and Methods

### 2.1. Betta Fish Sampling and Extraction of Microbial DNA and Measurements of Physiological Parameters

16-week-old male and female Siamese fighting fish were obtained from a betta fish farm in Chiang Rai, Thailand (20°2′29.795″ N, 99°53′0.57″ E). The fish were properly managed under the guidance of a Siamese fighting fish specialist. The fish were then mated to produce juveniles. They were transferred to a glass tank measuring 60 cm × 30 cm × 30 cm and kept in water using a flow-through system (Figure S1). The water quality was maintained as follows: dissolved oxygen (DO) 6.5–8.0 mg/L, water temperature (Tm) 23 to 25 °C, and pH 6 to 7. All the parameters listed above were within the acceptable range for fish growth [15,16]. All juveniles were together in a tank and fed a diet containing *Moina macrocopa* once daily at 3% of body weight (BW). Individual male and female fish were kept separately in individual tanks from 16 weeks of age and grew to 24 weeks. The basic characteristics of host Siamese fighting fish are provided in Table 1. Individual fish were sampled on the final day of each trial at 8, 16, and 24 weeks for standard length and/or width measurements (cm). Precision weighing equipment was also used to determine the standard weight of each fish (gram). As previously documented [1,11,17], the fish fin (pectoral fin and ventral fin) was used to determine the age and gender of betta fish.

**Table 1.** Basic characteristics of Siamese fighting fish used in this study.

| Measurement | Young | Female | | Male | | *p*-Value |
|---|---|---|---|---|---|---|
| | B8W | BF16W | BF24W | BM16W | BM24W | |
| Width (cm) | 0.22 ± 0.03 [a] | 0.81 ± 0.06 [a,b] | 0.69 ± 0.05 [a,b] | 0.86 ± 0.04 [b] | 0.85 ± 0.00 [a,b] | 0.02 [k] |
| Length (cm) | 0.87 ± 0.06 [a] | 3.29 ± 0.21 [a,b] | 3.19 ± 0.10 [a,b] | 3.82 ± 0.12 [b] | 3.61 ± 0.05 [a,b] | 0.01 [k] |
| Weight (g) | 0.09 ± 0.01 [d] | 0.39 ± 0.07 [b,c] | 0.31 ± 0.04 [c] | 0.63 ± 0.10 [a] | 0.50 ± 0.02 [a,b] | <0.0001 [w] |

Compact letters ([a], [b], [c] and [d]) in the same row indicate significant differences in pairwise comparisons determined by the Dunn's tests or Tukey's HSD test following a significant the [k] Kruskal–Wallis test or [w] one-way ANOVA test, respectively (*q* < 0.05, multiple testing corrections using the Benjamini-Hochberg method). B8W, 8-week-old Siamese fighting fish; BM16W, 16-week-old male Siamese fighting fish; BF16W, 16-week-old female Siamese fighting fish; BM24W, 24-week-old male Siamese fighting fish; BF24W, 24-week-old female Siamese fighting fish.

## *2.2. Extraction of Microbial Genomic DNA*

A total of 45 samples were collected for profiling the gut microbiota at weeks 8, 16, and 24 and divided into five groups based on age and gender as follows: (1) 8-week-old group (B8W, *n* = 9), (2) 16-week-old male group (BM16W, *n* = 9), 16-week-old female group (BF16W, *n* = 9), (4) 24-week-old male group (BM24W, *n* = 9), and 24-week-old female (BF24W, *n* = 9). Samples taken from 8-week-old fish were chosen to represent the juvenile tract as their morphology is still developing [1,2]. Individual fish were then randomly selected and pooled into three subgroups for subsequent analyses (B8W1-B8W3, BM16W1-BM16W3, BF16W1-BF16W3, BM24W1-BM24W3, and BF24W1-BF24W3). None of the fish in this study showed any external signs of disease. Bacterial genomic DNA was extracted from 200–400 mg of the entire gastrointestinal (GI) tract of fish using the Qiagen DNA stool mini kit (Qiagen, Hilden, Germany), according to the manufacturer's instructions with the following modifications: gut tissues were homogenized using 0.5 mm zirconia beads. The extracted DNA was stored at −80 °C until further molecular analysis.

## *2.3. 16S rRNA Gene Amplicon Sequencing*

The V3-V4 hypervariable region of the 16S rRNA gene was amplified using the primers 341F (5′-CCTAYGGGRBGCASCAG-3′) and 806R (5′-GGACTACNNGGGTATCTAAT-3′) with the barcode. The PCR reactions were performed using Phusion® High-Fidelity PCR Master Mix (New England Biolabs). The PCR products were analyzed using 2% agarose gel and purified with Qiagen Gel Extraction Kit (Qiagen, Germany). Then, sequencing libraries were generated using the NEBNext® Ultra™ IIDNA Library Prep Kit (Cat No. E7645). The library quality was verified on a Qubit@ 2.0 Fluorometer (Thermo Scientific) and Agilent Bioanalyzer 2100 system. 250 bp paired-end reads were obtained on the Illumina NovaSeq platform (Illumina, San Diego, CA, USA) at Novogene (Hong Kong).

## *2.4. Bioinformatics and Statistical Analysis*

### 2.4.1. Taxonomic Annotation of Amplicon Sequence Variants (ASVs)

The raw paired-end reads were primer- and barcode-trimmed and were merged using FLASH (version 1.2.11, http://ccb.jhu.edu/software/FLASH/, accessed on 8 June 2022) [18]. The fastp (version 0.20.0) software was used to generate high-quality clean tags. The chimera sequences were detected by comparing the clean tags with the SILVA database using Vsearch (version 2.15.0) and filtered out to obtain the effective tags [19]. The sequences were analyzed with the QIIME2 software (version QIIME2-202006). Amplicon sequence variants (ASVs) were assembled using the DADA2 pipeline [20]. The ASVs with an abundance of less than five were removed from the dataset [21]. The representative sequences were taxonomically classified against the SILVA Database. Host (contamination) sequences (e.g., mitochondria) were also removed from annotated reads. New names have been given to the phyla of prokaryotes, including Bacteroidota, Fusobacteriota, Verrucomicrobiota, Spirochaetota, Actinobacteriota, Acidobacteriota, which were previously recognized as Bacteroidetes, Fusobacteria, Verrucomicrobia, Spirochaetes, Actinobacteria, and Acidobacteria, respectively. The effective reads of 16S rRNA genes of Siamese

fighting fish gut microbiome obtained in this study have been deposited at the NCBI SRA database under the Bioproject accession number PRJNA873232 (BioSample accession numbers SAMN30492725–SAMN30492739).

### 2.4.2. Alpha and Beta Diversity Analyses

Samples were rarefied to a minimum of 79,055 reads per sample. Alpha and beta diversity analyses were conducted in the QIIME2 software. The principal coordinate analysis (PCoA) was generated using the R ade4 package. Plots were visualized using the R ggplot2 package in R version 2.15.3. Ternary plot analysis for the differences in the dominant species of the three groups was performed using the R vcd package. Variations in community structure and composition based on weighted and unweighted Unifrac distance matrices between groups were evaluated using the adonis function in the QIIME2 software. Differences in composition between groups were determined using T-test analysis embedded in the R software (version 3.5.3). Differentially abundant gut microbiota was analyzed using LEfSe (linear discriminant analysis (LDA) Effect Size) analysis [22].

### 2.4.3. Network Analysis

For co-occurrence network analysis, ASVs with mean relative abundance above 0.1% across all samples were chosen to investigate microbial interaction networks. The CoNet ensemble app in Cytoscape version 3.9.1 was used to construct networks [23]. The minimum row occurrence was set to 1 for all samples, as recommended by the program. The bacterial associations were identified using the Spearman correlation with a 0.8 threshold. Edges were verified through 1000 permutations. Edges were validated using 1000 permutations. After Benjamini-Hochberg correction, edges with a *p*-value less than 0.05 (*q*-value) were visualized using the R package igraph version 1.2.6 [24].

### 2.4.4. Functional Prediction of Gut Microbiota

Further, the function profiles of microbial communities were predicted using the PICRUSt2 software (version 2.1.2-b) according to KEGG Orthology (KO). Each annotated gene was represented as a K number (K). A heatmap of predicted Siamese fighting fish gut microbiota function at levels 2 and 3 was plotted using the R ComplexHeatmap package (version 2.5.4) [25] in R version 4.0.3 [26]. Dunn's test (following a significant Kruskal–Wallis test), the Wilcoxon rank sum test, or *t*-test were used to assess differences in the mean relative abundance of annotated gene functions between groups. A significant association between the relative abundance of predicted genes (Ks) and gut microbiota at the genus level was identified by hierarchical all-against-all association (HAIIA) [27] in Python (version 3.8.13).

## 3. Results

### 3.1. Microbial Diversity of the Siamese Fighting Fish Intestinal Tract

Deep sequencing of bacterial 16S rRNA genes from five distinct samples was used to explore the diversity of the gut microbiota. The number of ASVs ranged from 96,403 to 264,371 (Supplementary File S1). For assessing diversity, five indices were used: Chao1, observed species, and Shannon (Figure 1a). The analysis showed that there was no difference in diversity among groups. However, based on the above three indices, a decreasing trend was observed in the BM24W and BF24W groups.

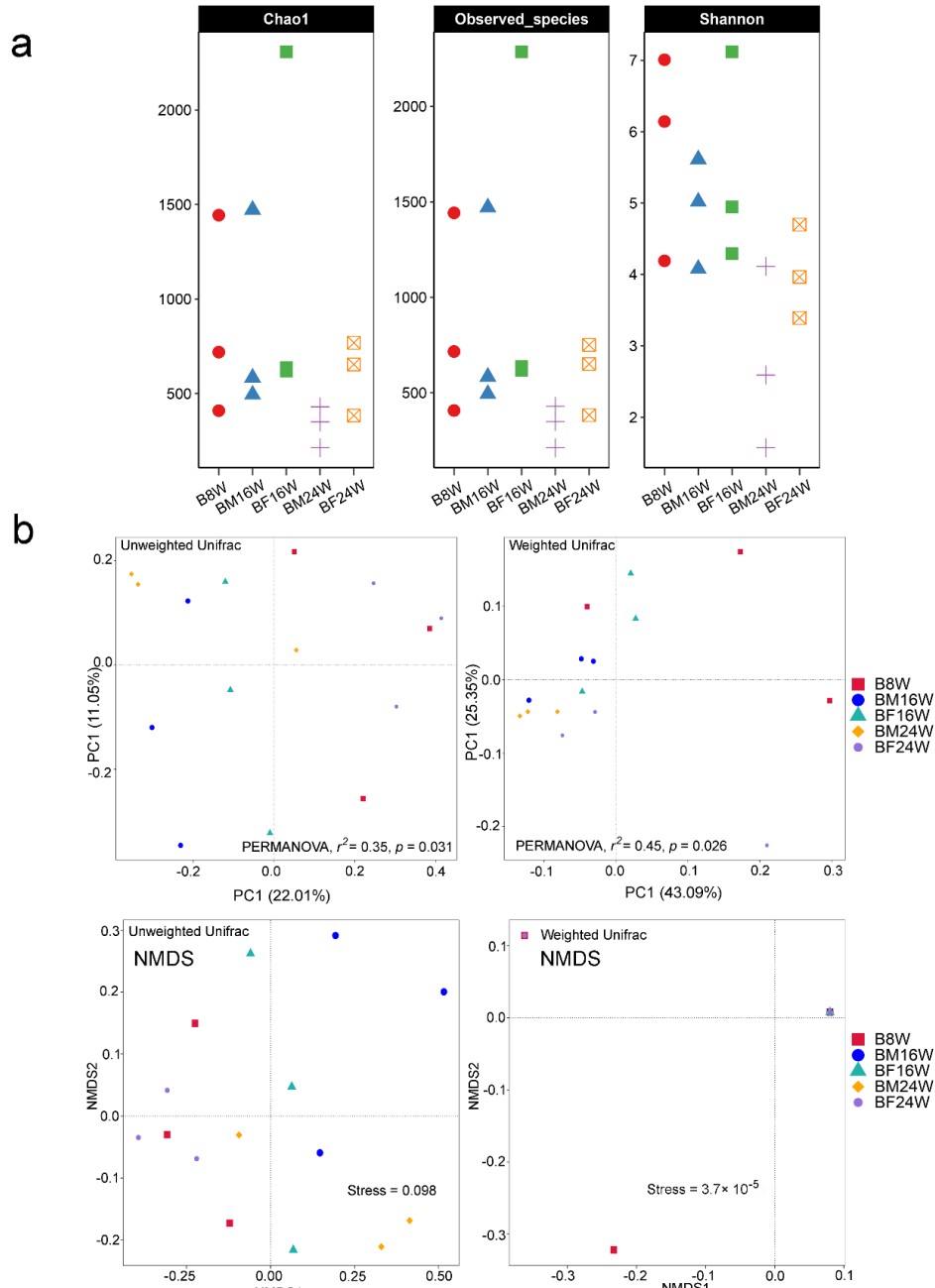

**Figure 1.** Gut microbiota diversity in Siamese fighting fish across age and gender groups. (**a**) Boxplots of alpha diversity of Siamese fighting fish gut microbiota across age and gender groups. Alpha diversity index was calculated based on Chao1, observed species, and Shannon indices. (**b**) Siamese fighting fish gut microbial community structure and composition across age and gender groups. PCoA plots of unweighted and weighted UniFrac distances, respectively. A difference in microbial communities across groups was determined by PERMANOVA. NMDS plots of gut microbiota based on unweighted and weighted UniFrac distances, respectively. The closer the stress value is to 0, the better the original data points are represented in the reduced dimensions. The distances between the sample points represent the dissimilarity of their microbiomes. B8W, 8-week-old Siamese fighting fish; BM16W, 16-week-old male Siamese fighting fish; BF16W, 16-week-old female Siamese fighting fish; BM24W, 24-week-old male Siamese fighting fish; BF24W, 24-week-old female Siamese fighting fish.

We then examined beta diversity using unweighted and weighted UniFrac distance matrices to determine the overall gut microbiota community structure and composition. PCoA of unweighted Unifrac (Figure 1b) revealed variation in microbial communities among groups, whereas PCoA of weighted Unifrac showed a close distance between adult Siamese fighting fish clusters. PERMANOVA also indicated significant differences in gut microbiota community structure (unweighted Unifrac, $p = 0.031$) and composition (weighted Unifrac, $p = 0.026$). Furthermore, a non-metric multidimensional scaling (NMDS) plot based on weighted Unifrac showed that the gut microbiota composition of the adult group was strongly clustered (stress value < 0.0001), while the group of 8 weeks old was dispersed (Figure 1b). Altogether, these results indicated that the gut microbiota of Siamese fighting fish of different ages and gender was more diversified in structure rather than composition.

### 3.2. The Composition of Gut Microbiota in Young and Adult Siamese Fighting Fish

Proteobacteria, Firmicutes, Bacteroidota, Fusobacteriota, Verrucomicrobiota, Campylobacterota, Spirochaetota, Actinobacteriota, Synergistota, and Acidobacteriota were among the top 10 phyla found in Siamese fighting fish gut samples (Figure 2a). Proteobacteria was the most prevalent phylum (more than 80% of total ASVs) in all samples, regardless of host age or gender. Differences in other phyla were observed between groups. Bacteroidota significantly dominated the gut of the young samples compared to that of the BM24W ($p = 0.04$) group. Fusobacteriota were more abundant in the guts of 16-week-old females than 24-week-old females ($p = 0.03$). Acidobacteria were less abundant in BF24W than in B8W ($p = 0.03$) and BM24W ($p = 0.01$). However, a significant difference was not preserved after multiple testing corrections.

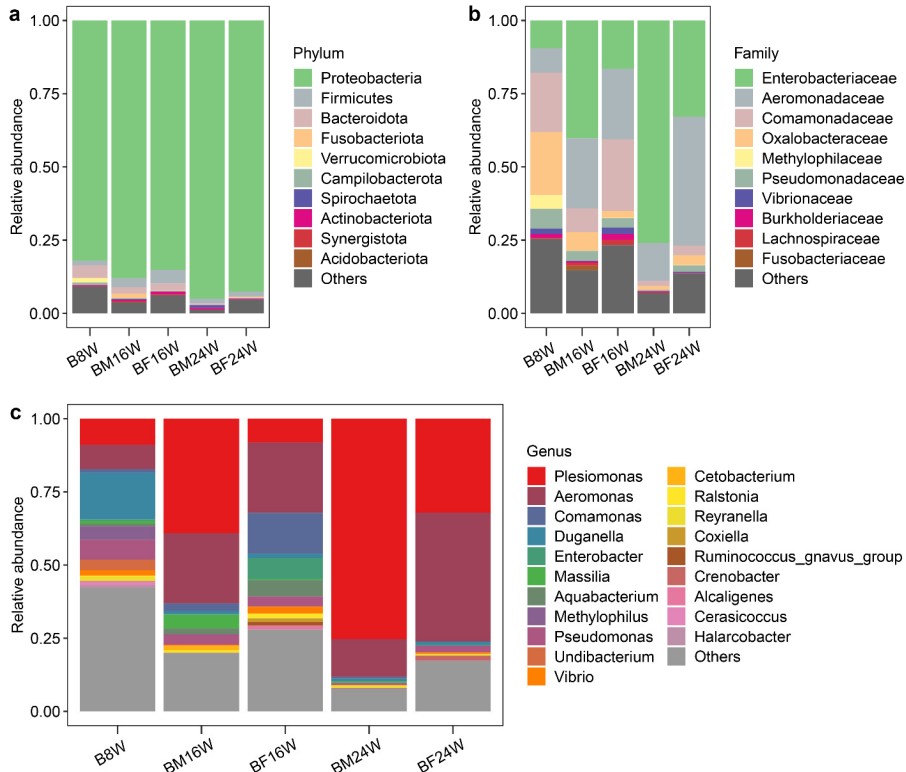

**Figure 2.** The top abundant gut microbiota in Siamese fighting fish across age and gender groups. (**a**,**b**) Bar plots of the average relative abundance of the top ten phyla and families. (**c**) A bar plot of the average relative abundance of the top 20 genera. Others are the low relative abundance of ASVs outside the top taxa. B8W, 8-week-old Siamese fighting fish; BM16W, 16-week-old male Siamese fighting fish; BF16W, 16-week-old female Siamese fighting fish; BM24W, 24-week-old male Siamese fighting fish; BF24W, 24-week-old female Siamese fighting fish.

At the family level, the most abundant taxon in all samples was *Enterobacteriaceae* (accounting for 9–76% of total ASVs), followed by *Aeromonadaceae* and *Comamonadaceae* (Figure 2b). In the BM24W samples, the level of Enterobacteriaceae was also greater (*p* = 0.01) than in the B8W group. *Pseudomonadaceae* dominated the gut of young Siamese fighting fish compared to 24-week-old males (*p* = 0.02) and females (*p* = 0.03) (Figure 3a). Interestingly, *Holophagaceae* were virtually absent from the BF24W samples. However, after multiple testing corrections, no significant difference was preserved at the phylum and family levels.

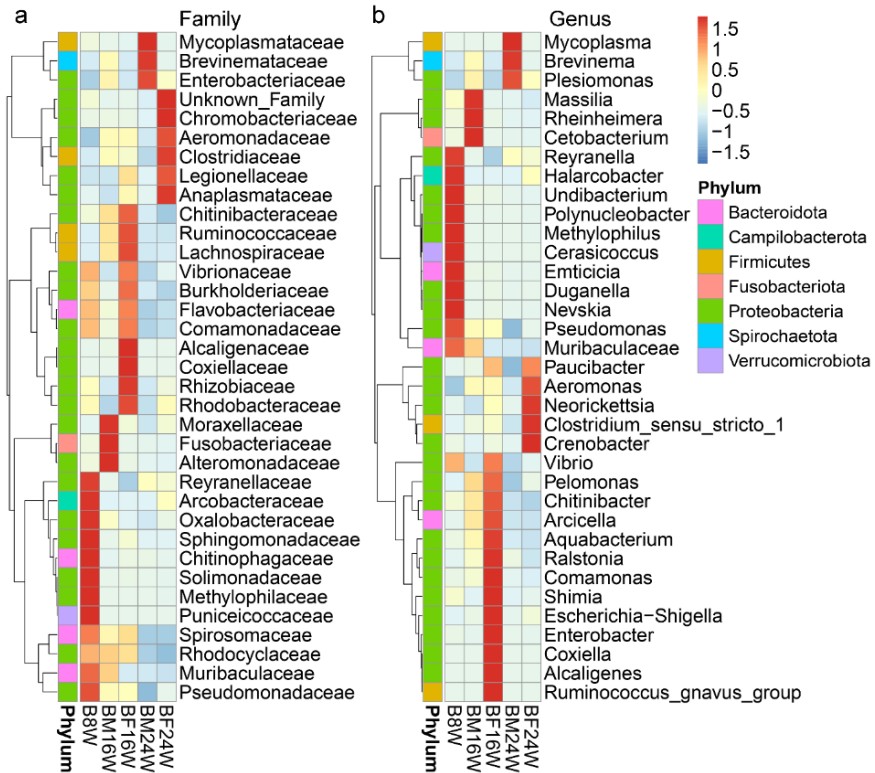

**Figure 3.** Heatmap of hierarchical clustering of top 35 bacterial families (**a**) and genera (**b**) across age and gender groups of Siamese fighting fish. The left annotation represents gut microbiota at the phylum level. The block of gradients (high and low values) indicates the enrichment of gut microbiota across groups. B8W, 8-week-old Siamese fighting fish; BM16W, 16-week-old male Siamese fighting fish; BF16W, 16-week-old female Siamese fighting fish; BM24W, 24-week-old male Siamese fighting fish; BF24W, 24-week-old female Siamese fighting fish.

At the genus level, *Plesiomonas* and *Aeromonas* were the most prevalent genera in all groups (Figure 2c). The former was significantly higher in BM24W than in B8W (*q* = 0.02). *Pseudomonas* was least abundant in BM24W (less than 1%) and was significantly lower than that in B8W (*q* = 0.02). The abundance of *Comamonas* was also decreased in the 24-week-old male and female samples (less than 1% on average) and was lower than in the younger group (B8W vs. BF24W, *q* < 0.01). *Aquabacterium* displayed a similar pattern, with its abundance being lower in 24-week-old adults than in the B8W group (B8W vs. BM24W, *q* < 0.01). Furthermore, the LEfSe analysis revealed that the above genera were notably associated with different age groups within the same gender. *Pseudomonas* was significantly enriched in the gut of young aged Siamese fighting fish (Figure 4b,c). This genus, along with *Comamonas*, was also more prevalent in 16-week-old male fish than in 24-week-old male fish (Figure 4e). The latter genus and *Aquabacterium* were more abundant in 16-week-old female adults than in 24-week-old female adults (Figure 4f). Collectively, our findings demonstrated that the developmental stages of Siamese fighting fish, from

young to adult, had an impact on the composition of gut microbiota, and gender became a determinant as the host matured.

**Figure 4.** LEfSe analysis of the relative abundances of gut microbiota in Siamese fighting fish across age and gender groups. Adult groups of the same gender were compared to the young group (B8W). (**a**–**g**) Differentially abundant taxa across groups with LDA score > 4.0. LEfSe analysis showed no bacteria taxa enriched in B8W, BM16W, or BF16F. B8W, 8-week-old Siamese fighting fish; BM16W, 16-week-old male Siamese fighting fish; BF16W, 16-week-old female Siamese fighting fish; BM24W, 24-week-old male Siamese fighting fish; BF24W, 24-week-old female Siamese fighting fish.

### 3.3. Gender Had a Significant Influence on the Structure of the Microbial Community Network of Siamese Fighting Fish

According to the co-occurrence microbial networks, significant associations between the gut microbiota at the genus level were observed across groups of Siamese fighting fish ($q < 0.0001$). The microbial community members mainly formed mutually exclusive relationships regardless of the host's age or gender (Supplementary File S2) and were not strongly connected to the neighborhood (Table S1). Notably, Proteobacteria accounted for a large proportion of all network connections. A similar pattern of microbial associations

was observed in the young and adult male groups. For instance, The B8W, BM16W, and BM24W communities showed strong negative associations of *Plesiomonas* with *Duganella* and *Rurimicrobium* (*rho* = −1, *q* < 0.0001) (Figure 5, Figures S2 and S3). In male samples (BM16W and BM24W), *Plesiomonas* were negatively associated with *Comamonas* and *Ralstonia* (*rho* = −1, *q* < 0.000). The female groups shared a negative relationship pattern of *Enterobacter* with several bacteria, including *Aquabacterium*, *Paucibacter*, and *Pelomonas*, all belonging to Proteobacteria (*rho* = −1, *q* < 0.000). Furthermore, *Plesiomonas* was the most active bacteria in the male network, with its degree being highest in the BM24W samples (negative degree = 32, betweenness = 1015.75), whereas a few of its relationships were also observed in the young and female fish. This indicated that *Plesiomonas* might play a major role in the mutual exclusion networks of male Siamese fighting fish.

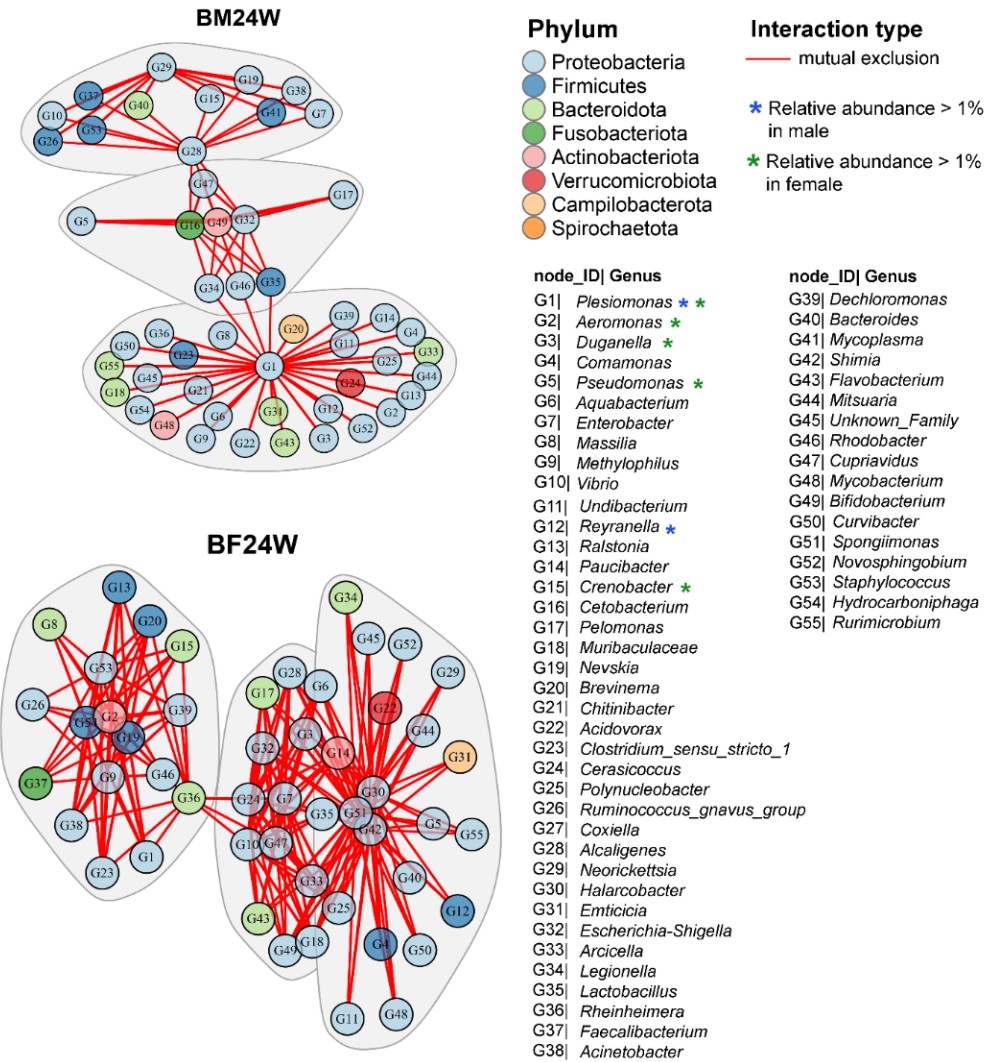

**Figure 5.** Microbiota co-occurrence network graphs at the genus level in 24-week-old Siamese fighting fish. The Spearman correlation coefficient was used to calculate network interactions between gut microbiota. The cutoff for a significant correlation was 0.8 (*q* < 0.05, multiple testing corrections using the Benjamini-Hochberg method). The presence of red edges indicates mutual exclusion (negative correlation). The colors of the nodes represent the phylum level. Gray shades are used to color each module. Bacterial genera are represented by node IDs. The minimum row occurrence was set to one (suggested by the CoNet app). BM24W, 24-week-old male Siamese fighting fish; BF24W, 24-week-old female Siamese fighting fish.

*3.4. Associations between Siamese Fighting Fish Gut Microbiota and Predicted Function*

Siamese fighting fish gut microbiota functional prediction based on KEGG Orthology (KO) revealed a total of 7870 identified KEGG genes, corresponding to 51 KEGG level 2 pathways and 442 KEGG level 3 pathways (Supplementary File S3). The most abundant level 2 and level 3 pathways were protein families: metabolism and enzymes, accounting for 31% and 25% of the total predicted pathways, respectively (Figure S4).

Comparisons of bacterial genes based on identified KEGG genes revealed that only significant differences in the relative abundance of these KEGG genes were observed between the young group and 24 weeks old, as well as between 24 weeks old male and female Siamese fighting fish ($q < 0.05$) (Supplementary File S4). No difference was found when comparing the groups of 8 and 16 weeks old. The functional orthologs of all KEGG genes were then categorized based on KEGG pathway levels. Regarding the top 30 pathways, we observed a similar distribution pattern across groups, with enzymes and transporters being the most enriched pathways at level 3 that represented bacterial gene function in Siamese fighting fish gut microbiota (Figure 6). When considering all groups, gender, and age, comparisons of the mean relative abundance of the top 30 pathways at level 2 between groups revealed that only significant differences were detected between the B8W and BM24W groups ($q < 0.05$). While exosome was more enriched in the B8W group, other eight pathways (relevant to enzymes, purine metabolism, pyrimidine metabolism, ribosome biogenesis, transfer RNA biogenesis, peptidoglycan biosynthesis, and degradation proteins, chromosome and associated proteins, and amino sugar and nucleotide sugar metabolism) were significantly higher in the BM24W group. These pathways were associated with the metabolism, gene and protein pathways at level 1.

We further determined the association between predicted function and Siamese fighting fish gut microbiota using hierarchical all-against-all association (HAIIA) to better understand the relationship between them. The analysis showed that gut microbiota at the genus level exhibited a strong correlation ($-1$ or 1, spearman, $q < 0.0001$) with the top 50 KEGG genes in all groups (Supplementary File S5). Moreover, an intriguing pattern was observed when we focused on the three dominant genera belonging to the Proteobacteria phylum, including *Plesiomonas*, *Comamonas*, and *Pseudomonas*. The number of unique associations was greater than shared associations based on Siamese fighting fish maturation stages (Figures S5a–S7a and Figure 7a). In the young group (B8W), *Plesiomonas* exhibited a distinct negative correlation with 33 KEGG genes involved in 28 pathways (e.g., protein families-signaling and cellular processes, protein families-metabolism, carbohydrate metabolism, and lipid metabolism), whereas *Comamonas* and *Pseudomona* were positively correlated with 9 KEGG genes pathways, the majority of which were involved in protein families related to signaling and cellular processes as well as the quorum sensing-related gene. In adults, the association of *Plesiomonas* with KEGG genes consistently contrasted with *Comamonas* or *Pseudomonas* as observed in the BM16W, BF16W, and BM24W groups (Figure 7c–e, Figures S5d, S6d–e and S7d). For instance, in the BM16W group. *Plesiomonas* was negatively correlated with K00059 (3-oxoacyl-[acyl-carrier protein] reductase), while *Comamonas* exhibited a positive relationship with this KEGG gene involved in several pathways, including lipid metabolism. In the gut of 16-week-old Siamese fighting fish, *Pseudomonas* was positively correlated with K00799 (glutathione S-transferase), which was involved in glutathione metabolism, whereas *Plesiomonas* was negatively correlated with this gene. Furthermore, in the BM24W group, *Pseudomonas* ($-$) and *Plesiomonas* ($+$) showed a contrast correlation with K02003 (ABC transport system ATP-binding protein) and K02004 (ABC transport system permease protein), while *Comamonas* displayed a negative relationship with the quorum sensing-related genes (K02034). *Pseudomonas* also showed a positive relationship with K01992 (ABC-2 type transport system permease protein) and K01091 (phosphoglycolate phosphatase) in the young Siamese fighting fish, whereas in male adults (BM16W and BM24W), this genus displayed a negative relationship with both KEGG genes relevant to protein families-signaling and cellular processes and carbohydrate metabolism, respectively (Figure 7c–e). The latter KEGG gene, K01091, was significantly

lower in BM24W than in B8W ($q < 0.01$). In addition, *Aeromonas*, the second most abundant genus in Siamese fighting fish, only had a negative association with functional genes in B8W, such as protein families-signaling and cellular processes, protein families-metabolism, and carbohydrate metabolism. Its association with KEGG genes, however, decreased with host age and was lowest in 24-week-old female fish (only three negative associations). Surprisingly, *Aquabacterium* had a different profile than *Aeromonas*, with the degree of its association with KEGG genes (e.g., protein families-signaling and cellular processes, protein families-metabolism, and cellular community) shifting towards positive connection in adult fish and being highest in 24-week-old Siamese fighting fish. We postulated that as the host developed, gut microbiota function might be switched to compensate for one another's activity. Overall, our results indicated that age and gender influenced the functional profile of the gut microbiota in Siamese fighting fish.

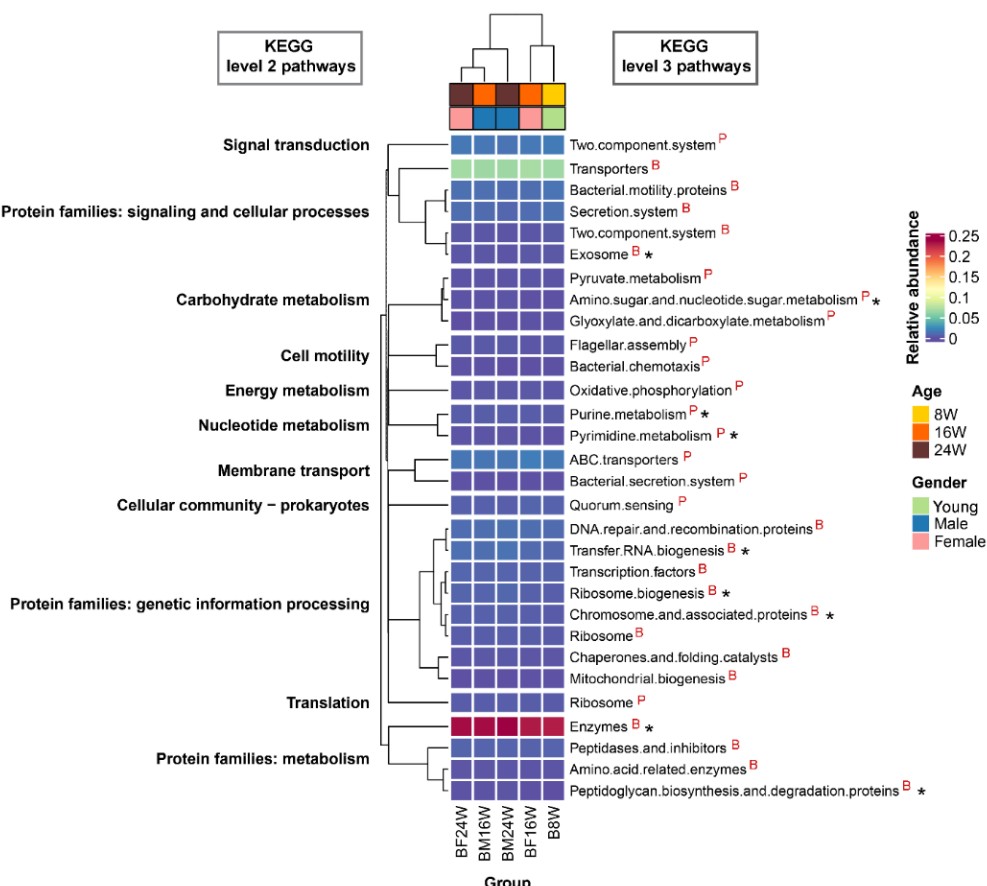

**Figure 6.** Hierarchical clustering of predicted Siamese fighting fish gut microbiota function across age and gender groups based on Bray-Curtis dissimilarities. The heatmap represents the relative abundances of microbial gene functions identified at the KEGG level 3 pathway. Rows are split according to identified KEGG level 2 pathways. It should be noted that heatmap construction excluded 1047 KEGG genes related to orthologs, modules, and networks that were not included in 'Pathway or Brite.' An asterisk (*) indicates significant differences across groups ($p < 0.05$). Top annotations refer to the age and gender of Siamese fighting fish. KEGG database classification: P, Pathway maps; B, Brite hierarchies and tables. B8W, 8-week-old Siamese fighting fish; BM16W, 16-week-old male Siamese fighting fish; BF16W, 16-week-old female Siamese fighting fish; BM24W, 24-week-old male Siamese fighting fish; BF24W, 24-week-old female Siamese fighting fish.

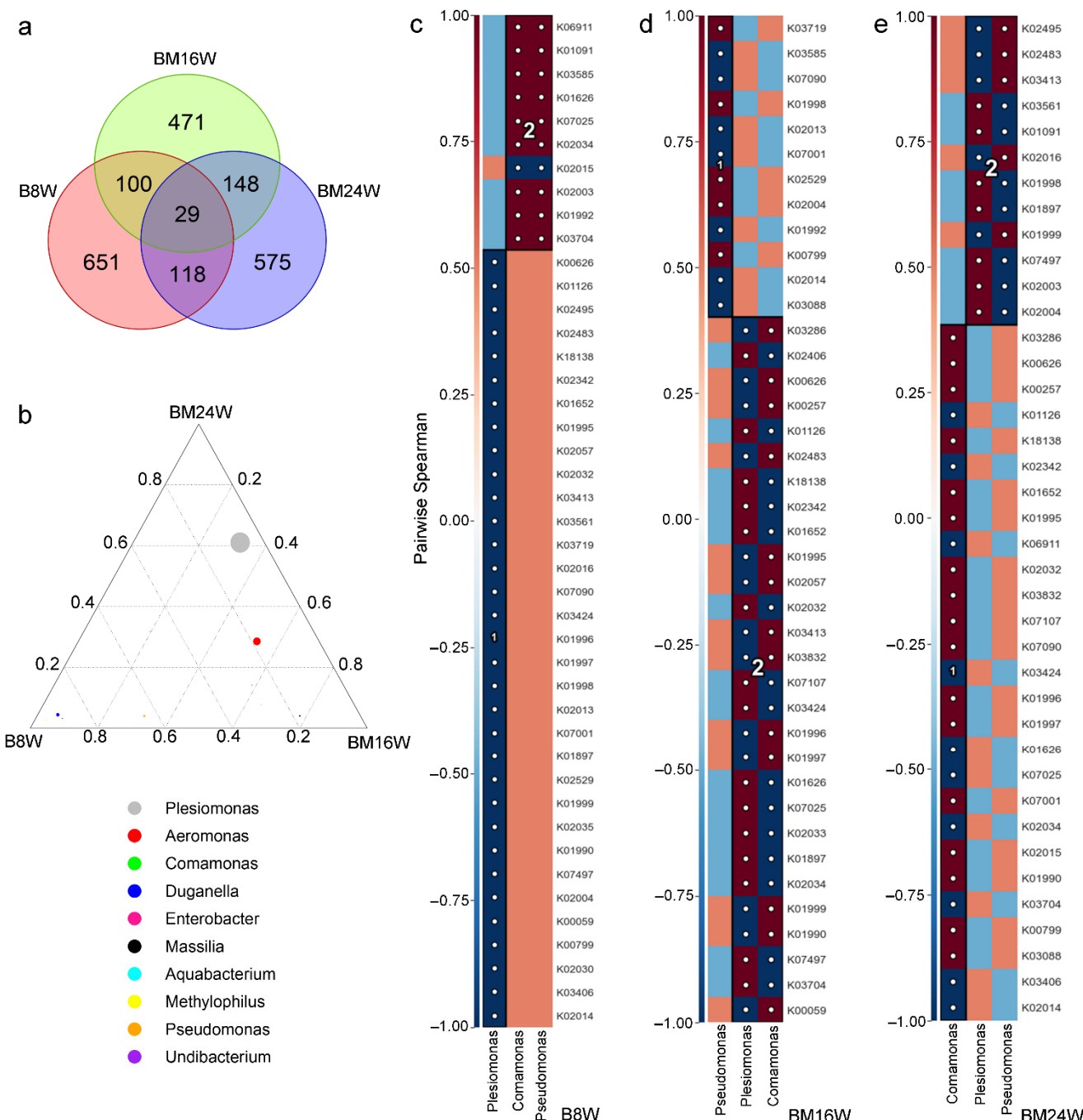

**Figure 7.** Association between Siamese fighting fish gut microbiota and predicted function. (**a**) Numbers of shared and unique associations between young and male adult samples were identified by hierarchical all-against-all association (HAIIA). (**b**) A ternary plot of the top 10 genera across groups. The size of circles is proportional to the relative abundance. The closer the circle is to the vertices, the more abundant a certain genus is in a sample. (**c–e**) Heatmaps representing a significant relationship between the dominant Siamese fighting fish gut microbiota and top 50 K numbers in terms of relative abundance (*q* < 0.0001) in the B8W, BM16W, and BM24W groups (respectively). B8W, 8-week-old Siamese fighting fish; BM16W, 16-week-old male Siamese fighting fish; BM24W, 24-week-old male Siamese fighting fish.

## 4. Discussion

In this study, bacterial 16s rRNA sequencing was conducted to investigate gut microbiome features related to fish age and gender. Alpha diversity indices indicated a noticeable decreasing trend in the gut microbiome during aging, regardless of gender; however, differences between groups were not significant. This could imply that as Siamese fighting fish age, their bacterial diversity declines. Our results were in contrast to a previous study on Southern catfish (*Silurus meridionalis*), which found increasing microbial diversity as the host matured; however, both studies agreed that community composition varied with host age [28]. These findings suggested that gut bacterial diversity in fish is shaped by age [10,29,30]; however, this may vary depending on the fish species.

PERMANOVA and NMDS were used to perform beta diversity analysis. These revealed significantly different variations of the bacterial community in terms of structure rather than composition. Proteobacteria was the most prevalent phylum in all groups. Previous investigations on freshwater fish yielded similar results. This phylum is the major group of bacteria found in fish [31,32], and their relative abundance differences can be explained by the partial projection of the vast diversity of Proteobacteria associated with aquatic habitats or by the host itself [33]. Moreover, most members of this phylum are known protease producers able to digest protein, especially in carnivorous fish [34]. Hence, the high level of Proteobacteria in the Siamese fighting fish might be attributed to a high-protein source diet, like *Moina*. Bacteroidota was the second most dominant phylum in young fish, which significantly decreased in adults, particularly in males. Bacteroidota has been reported as the dominant flora in herbivorous fishes [35]. On the other hand, betta fish is a carnivore that naturally consumes insects and insect larvae [1]. As diet is one of the important factors in defining gut microbiota in fish [33], this phylum may be innate bacteria whose abundance decreases in response to the nutritional status of Siamese fighting fish fed on *Moina* (a crustacean) in this study. It should be noted that overrepresented members of Proteobacteria (such as *Vibrio*, *Aeromonas*, and *Shewanella*) could potentially cause diseases in other fish [36] and even in humans [37], which raises concerns [38]. Whether these microbes can spread throughout the food chains remains to be investigated.

The environment specifies gut microbial composition in fish [33,39]. Our Siamese fighting fish growing in freshwater, particularly adults, mainly harbored the members of *Gammaproteobacteria*, *Plesiomonas*, and *Aeromonas*. This finding is consistent with other studies reporting that these bacteria are common colonizers in the gut of freshwater fish [40,41]. Notably, *Aeromonas* spp. are known as cellulase and amylase producers [7,42,43]. These bacteria could be the key taxa involved in carbohydrate metabolism promoting the growth of Siamese fighting fish. However, the lower proportion of the above dominant bacteria in young fish that are known to cause opportunistic infections (*Plesiomonas* and *Aeromonas*) [41,44] may also be influenced by the host immune system. Since young fish may not have fully developed immune systems, this bacterial group may be controlled by host immune responses.

Mutual exclusion appeared to be conserved in Siamese fighting fish, but pairwise associations between individual members changed as the host developed, with Proteobacteria dominating all networks. This bacterial group had the most associations with cichlid fishes [45]. According to the authors, the host gut environment may shape the ecological relationships of fish gut communities (e.g., physiology and immune system).

Several factors, including age and stage of maturity, can influence the community of bacteria in the fish gut, which in turn can influence how bacteria function in the gut [10]. In this study, host age and gender appeared to play a dominant role in shaping the community function in the gut of Siamese fighting fish. The results of PICRUSt2 revealed variations in predicted functional genes of gut bacteria assembly. The gut microbial pathways related to metabolism were more enriched in 24-week-old males than in 8-week-old samples. This implies that intestinal metabolic activity increases as the host ages. A greater abundance of bacterial genes involved in carbohydrate metabolism in male adults may indicate that more energy is supplied to the host fish, thereby promoting growth, as the gut microbiota makes an important contribution to digestion which has an impact on host health [46].

The size and strength required for aggressive behavior in adult males may also engage in this phenomenon. Furthermore, the dominant gut microbiota in young Siamese fighting fish exhibited a distinct association with functional genes. The negative association of *Plesiomonas* and the positive association of *Comamonas* and *Pseudomonas* with many genes related to metabolism, signaling, and cellular processes might imply that these bacteria play different roles in the young fish gut for maintaining host health. The latter two bacteria were found to be more enriched in the skin mucus of healthy catfish (*Heteropneustes fossilis*) than in diseased catfish [47]. Hence, it was suggested that these microbes could serve as potential probiotics in aquaculture. Nonetheless, the pathogenicity of a genus like *Pseudomonas* may vary depending on the strain, as the genus is associated with both healthy and diseased hosts [48].

The association pattern of gut microbiota with KEGG genes related to ABC transporters (K02003 and K02004) and quorum sensing (K02034 and K02035) shifted as the host developed. Bacterial ABC transporters play a variety of roles, from nutrient uptake to bacterial pathogenicity, all of which are required for survival [49]. Quorum sensing (QS) is also an essential process for bacterial communication, which helps balance the community as well as reduce bacterial virulence in the host gut [50]. We observed a positive relationship between the aforementioned genes and *Comamonas* or *Pseudomonas* in the young Siamese fighting fish but a negative relationship in adult samples. As the physiological state of the host body was not fully developed, these suggested that the gut microbiota might strengthen their viability through these functional processes. Furthermore, as the abundance of *Pseudomonas* was enriched in the gut of 8-week-old fish, a positive association of this genus with functional genes related to signaling and cellular processes and carbohydrate metabolism hinted that it might support the growth of the young Siamese fighting fish. A recent study conducted on freshwater fish (*Aplodinotus grunniens*) revealed an increased abundance of *Pseudomonas* following feed domestication, which enhanced ingestion and growth performance [32]. Future research into the biological function of Proteobacteria in mediating the growth and development of Siamese fighting fish is warranted, which could lead to gut microbiota manipulation for sustainable aquaculture.

There are limitations to this study, noted as follows. The number of animals used was limited due to ethical considerations. A pooling method may result in great variability within the group, which could obscure the gut microbiota community of individuals. In this regard, the gut samples could be independently analyzed to help reveal and evaluate the variation of the gut microbiota profile of each sample. The association between gut microbiota and the growth rate of the fish (from 8 to 24 weeks) could not be examined since the fish needed to be sacrificed at each time point. Profiling bacteria from water samples in future research might help to better understand the relationship between the fish gut microbiome and its aquatic environment. Furthermore, our sample is limited to *B. splendens* Regan. Future studies characterize the microbiome features of the entire betta fish population in order to gain insight into the association between host physiology and behavior and the gut microbiota in *Betta* species. Knowing these aspects may provide useful information on host-microbiome relationships, which could help improve fish health.

## 5. Conclusions

This study highlights the impact of Siamese fighting fish age and gender on the gut microbiota community. A decline in bacterial diversity was noted in 24-week-old adult fish. Proteobacteria dominate the gut of betta fish, with *Plesiomonas* and *Aeromonas* being the most prevalent community members. Co-occurrence network analysis indicated mutual-exclusion relationships between the gut community members in all groups. *Plesiomonas* occupied the central position in the microbial interaction network of 24-week-old male fish and displayed a distinct association pattern with functional genes of *Pseudomonas* or *Comamonas* in adult samples. Changes in the functional profiles of members of the Proteobacteria between young and adult fish also suggest that the gut microbiota may serve different functions during host development. The results warrant future investigation

on the role of Proteobacteria using shotgun metagenomic methods to profile entire microbial genomes and to gain a thorough understanding of the impact of such complex interactions on Siamese fighting fish health.

**Supplementary Materials:** The following supporting information can be downloaded at: https://www.mdpi.com/article/10.3390/fishes7060347/s1, Figure S1: Study design and sample collection, Figure S2: Microbiota co-occurrence network graphs at the genus level in 8-week-old Siamese fighting fish. The Spearman correlation coefficient was used to calculate network interactions between gut microbiota. The cutoff for a significant correlation was 0.8 ($q < 0.05$, multiple testing corrections using the Benjamini-Hochberg method). The presence of red edges indicates mutual exclusion (negative correlation). The colors of the nodes represent the phylum level. Gray shades are used to color each module. Bacterial genera are represented by node IDs. The minimum row occurrence was set to one (suggested by the CoNet app). B8W, 8-week-old Siamese fighting fish, Figure S3: Microbiota co-occurrence network graphs at the genus level in 16-week-old Siamese fighting fish. The Spearman correlation coefficient was used to calculate network interactions between gut microbiota. The cutoff for a significant correlation was 0.8 ($q < 0.05$, multiple testing corrections using the Benjamini-Hochberg method). The presence of red edges indicates mutual exclusion (negative correlation). The colors of the nodes represent the phylum level. Gray shades are used to color each module. Bacterial genera are represented by node IDs. The minimum row occurrence was set to one (suggested by the CoNet app). BM16W, 16-week-old male Siamese fighting fish; BF16W, 16-week-old female Siamese fighting fish, Figure S4: Bar charts represent the proportions of top 10 predicted KEGG pathways of Siamese fighting fish gut microbiota at level 2 (a) and level 3 (b), Figure S5: Association between Siamese fighting fish gut microbiota and predicted function. (a) Numbers of shared and unique associations between young and female adult samples were identified by hierarchical all-against-all association (HAIIA). (b) A ternary plot of thetop 10 genera across groups. The size of circles is proportional to the relative abundance. The closer the circle is to the vertices, the more abundant a certain genus is in a sample. (c–e) Heatmaps representing a significant relationship between the dominant Siamese fighting fish gut microbiota and top 50 K numbers in terms of relative abundance ($q < 0.0001$) in the B8W, BF16W, and BF24W groups (respectively). B8W, 8-week-old Siamese fighting fish; BF16W, 16-week-old female Siamese fighting fish; BF24W, 24-week-old female Siamese fighting fish, Figure S6: Association between Siamese fighting fish gut microbiota and predicted function. (a) Numbers of shared and unique associations between young and 16-week-old samples were identified by hierarchical all-against-all association (HAIIA). (b) A ternary plot of the top 10 genera across groups. The size of circles is proportional to the relative abundance. The closer the circle is to the vertices, the more abundant a certain genus is in a sample. (c–e) Heatmaps representing a significant relationship between the dominant Siamese fighting fish gut microbiota and top 50 K numbers in terms of relative abundance ($q < 0.0001$) in the B8W, BF16W, and BF24W groups (respectively). B8W, 8-week-old Siamese fighting fish; BF16W, BM16W, 16-week-old male Siamese fighting fish; 16-week-old female Siamese fighting fish, Figure S7: Association between Siamese fighting fish gut microbiota and predicted function. (a) Numbers of shared and unique associations between young and 24-week-old samples were identified by hierarchical all-against-all association (HAIIA). (b) A ternary plot of the top 10 genera across groups. The size of circles is proportional to the relative abundance. The closer the circle is to the vertices, the more abundant a certain genus is in a sample. (c–e) Heatmaps representing a significant relationship between the dominant Siamese fighting fish gut microbiota and top 50 K numbers in terms of relative abundance ($q < 0.0001$) in the B8W, BF16W, and BF24W groups (respectively). B8W, 8-week-old Siamese fighting fish; BM24W, 24-week-old male Siamese fighting fish; BF24W, 24-week-old female Siamese fighting fish, Table S1: Global network topologies for microbial association networks in Siamese fighting fish, Supplementary File S1: Statistical results of data processing, the results of ASVs annotations, and the feature tables of each sample. Supplementary File S2: Co-occurrence network analysis of Siamese fighting fish gut microbiota, Supplementary File S3: The function profiles of Siamese fighting fish gut microbiota according to KEGG Orthology (KO), Supplementary File S4: Statistical results of comparisons of mean relative abundance of predicted function according to KEGG Orthology (KO) between groups, Supplementary File S5: A significant association between the relative abundance of predicted genes (Ks) and gut microbiota at the genus level identified by hierarchical all-against-all association (HAIIA).

**Author Contributions:** Conceptualization, methodology, S.P.; investigation, V.J. and S.P.; formal analysis, L.G.; visualization, L.G.; writing—original draft preparation, L.G., V.J., P.T. and S.P.; writing—review and editing, L.G., V.J., P.T. and S.P.; supervision, S.P.; funding acquisition, S.P. All authors have read and agreed to the published version of the manuscript.

**Funding:** This work was financially supported by the Thailand Science Research and Innovation (TSRI) (Grant Number: 652A01021) and Mae Fah Luang University.

**Institutional Review Board Statement:** The methods involving animals in this study were approved by the Animal Ethics Committee of Mae Fah Luang University under registry number: AR05/63. Animal use was restricted in order to adhere to the country's Buddhist morals.

**Data Availability Statement:** The following information was supplied regarding data availability: Sequences of the 16S rRNA gene of Siamese fighting fish gut microbiota are available at BioSample: SAMN30492725–SAMN30492739 (also available at BioProject: PRJNA873232). Corresponding sequences (Fastq format) are available at https://dataview.ncbi.nlm.nih.gov/object/PRJNA873232.

**Acknowledgments:** We would like to thank Channarong Wanthanjai for technical assistance and Eleni Gentekaki for constructive criticism of the manuscript.

**Conflicts of Interest:** The authors declare no conflict of interest. The funders had no role in study design, data collection, and analysis, decision to publish, or preparation of the manuscript.

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
