# Peer review of "Siamese Fighting Fish (Betta splendens Regan) Gut Microbiota Associated with Age and Gender"

_fishes, doi:10.3390/fishes7060347_

Round 1

Reviewer 1 Report

Comments

The study by Gruneck et al. explored the gut microbiota in Siamese fighting fish. They found that the age and gender influenced the microbial community of this kind of species. Generally speaking, the present study showed good results and the statistical analysis seemed reasonable. The manuscript could provide some valuable references for betta fish. However, the design of this research seems simple and I have several specific comments, as showed below:

Abstract

Good.

Introduction

1.      The author should strengthen the reasons why focused on this species and why research the effects of the gender and the different development stages.

2.      The author should explain clearly of the developmental stages of betta fish.

Materials and methods

1.      The author sampled 9 fishes (n=9) for per group. However, why only 3 points were showed in the supplementary data figure S2 and figure S3.

2.      Line 80, what samples were collected, the whole gut or part of the gut?

3.      What’s the reason the author selected the sample time 8, 16, 24 week? Why do not consider the earlier time such as 1 week, 2 week?

4.      The part of bioinformatics and statistical analysis here is too complex and should be concisely.

5.      The article should tested the accuracy of the 16S rRNA sequencing results according to the qPCR method.

Results

1.      The results of Alpha diversity index and PCoA analysis in the supplementary file should be located in the article.

Discussion

no comments

Reviewer 2 Report

The manuscript submitted by Gruneck et al. presents a study of the differences in the gut microbiota profile of healthy Siamese fighting fish during the young and adult life stages using 16S rRNA gene sequence analysis. The study revealed the overall bacterial profile between young and adult Siamese fighting fish by tests to the diversity of the microbial community. These findings provide implications for expanding our understanding of the role of the gut microbiota and its association with host factors.

I consider this article as well prepared, but I have some comments and suggestions for the authors that may improve its quality.

Abstract:

Please add some information about the experiment design. Furthermore, why is for Proteobacteria was the most predominant bacterial phylum in the gut of both young and adult fish in which the abundance of its members varied with age should also be clarified. Also the research findings are not clarified enough.

Introduction:

Please use the proper name and cite properly reference (according to MDPI guideline)

Please clarify why you set the experiment and explain the reason through related literatures.

Materials and methods:

You did not explain how you determined these parameters. Add to the experimental section the subtitle Analytical Analysis and explain all these analyses.

In my opinion the composition of bacteria in the water will determine the composition in the tissues, so I suggest you to add some information about these.

Results:

The results you presented is too basic, what the biologic function of the bacteria for the Siamese fighting fish growth and development parameter?

Diversity analysis: Unclearly described results, e. g. in Figure 3 we do not know whether the data refer to the sum of the results from LEfSe analysis of the relative abundances of gut microbiota in Siamese fighting fish across age and gender groups.

Discussion:

Please explain why? Have other studies obtained similar results? It is necessary to write which and what kind of bacteria can cause problems for fish and whether they can be transferred to the human body through the food chain.

In general, the obtained results are an interesting introduction to further research; I would suggest extending the scope of the research to other size range of Siamese fighting fish. A larger and more diversified amount of obtained data would then be valuable not only for local readers but also more globally.

Round 2

Reviewer 1 Report

The manuscript in present form can be accepted.

Reviewer 2 Report

Corrections to minor methodological errors and text editing should be considered.